# Recent Developments in the Role of Protein Tyrosine Phosphatase 1B (PTP1B) as a Regulator of Immune Cell Signalling in Health and Disease

**DOI:** 10.3390/ijms25137207

**Published:** 2024-06-29

**Authors:** Neve E. Read, Heather M. Wilson

**Affiliations:** Institute of Medical Sciences, School of Medicine, Medical Sciences and Nutrition, University of Aberdeen, Aberdeen AB25 2ZD, UK; n.read.21@abdn.ac.uk

**Keywords:** protein tyrosine phosphatase 1B (PTP1B), signalling, immune cells, inflammation, haematopoiesis, disease

## Abstract

Protein tyrosine phosphatase 1B (PTP1B) is a non-receptor tyrosine phosphatase best known for its role in regulating insulin and leptin signalling. Recently, knowledge on the role of PTP1B as a major regulator of multiple signalling pathways involved in cell growth, proliferation, viability and metabolism has expanded, and PTP1B is recognised as a therapeutic target in several human disorders, including diabetes, obesity, cardiovascular diseases and hematopoietic malignancies. The function of PTP1B in the immune system was largely overlooked until it was discovered that PTP1B negatively regulates the Janus kinase—a signal transducer and activator of the transcription (JAK/STAT) signalling pathway, which plays a significant role in modulating immune responses. PTP1B is now known to determine the magnitude of many signalling pathways that drive immune cell activation and function. As such, PTP1B inhibitors are being developed and tested in the context of inflammation and autoimmune diseases. Here, we provide an up-to-date summary of the molecular role of PTP1B in regulating immune cell function and how targeting its expression and/or activity has the potential to change the outcomes of immune-mediated and inflammatory disorders.

## 1. Introduction

Protein tyrosine phosphorylation regulates a broad range of fundamental cellular processes, including signalling, proliferation, migration and invasion [1,2]. The levels of phosphorylation are regulated by the concerted activities of protein tyrosine kinases (PTKs) and protein tyrosine phosphatases (PTPs). PTPs are a large family of enzymes, the vast majority of which can be classified as either classical PTPs targeting only tyrosine residues or VH-1-like PTPs, also known as dual-specificity PTPs, targeting both tyrosine and threonine/serine residues [3,4]. Classical PTPs have a highly conserved catalytic domain containing a cysteine residue responsible for dephosphorylating the substrate via nucleophilic attack. PTPs can be further divided into transmembrane receptor-like PTPs and non-transmembrane PTPs depending on the structural variation in the domains linked to core recognition motifs that determine their cellular localisation [3]. Transmembrane receptor-like PTPs (e.g., CD45) signal upon extracellular ligand binding and are involved in regulating cell–cell and cell–extracellular matrix (ECM) interactions, while non-transmembrane receptor PTPs contain a regulatory domain, which controls catalytic activity by interacting with the active site or by regulating substrate specificity [1,3,4,5]. Dysregulated tyrosine phosphorylation has implications for cellular metabolism and inflammation, giving rise to a range of human diseases, such as metabolic, haematopoietic, inflammatory and immune disorders along with cancer and neurodegenerative diseases [2]. As such, both tyrosine kinases and phosphatases are important therapeutic targets. One such target, protein tyrosine phosphatase 1B (PTP1B), is upregulated in several immune and inflammatory diseases, and the dysregulation of its expression and activity in immune cells is thought to contribute to pathology. Consequently, inhibition of its activity is currently an important drug target for cancer and cardiometabolic disorders. PTP1B, a non-transmembrane phosphatase, directly controls the signalling of several tyrosine kinase receptors such as epidermal growth factor receptor (EGFR), platelet-derived growth factor receptor (PDGFR), insulin receptors (IRs), leptin receptor and insulin-like growth factor receptor (IGFR), as well as indirectly through the dephosphorylation of receptor-associated kinases including PI3K/Akt/mTOR, RAS-MAPK and AMPK. These pathways are all involved in regulating cell growth, proliferation and immune cell function [6,7,8,9,10,11]. PTP1B also dephosphorylates the JAK2 and Tyk2 kinases and activates MAPKs and STATs, which are important immune regulators [12,13]. As PTP1B controls important signalling pathways relating to cell function, it has been considered a potential therapeutic target. The development of highly specific PTP1B inhibitors has, however, been challenging due to the highly conserved active site found across all PTPs that hampers the specificity of inhibitors. For example, PTP1B shares 72% structural similarity with T cell protein tyrosine phosphatase (TC-PTP), which can lead to PTP1B inhibitors having off-target effects [14,15]. Moreover, the catalytic site of PTP1B is positively charged, while catalytic site inhibitors are likely to be negatively charged or highly polar, thus preventing them from effectively crossing cell membranes to inhibit PTP1B. Several PTP1B inhibitors, including ertiprotafib, trodusquemine and ABBV-CLS-484, have, however, successfully reached phase I and II clinical trials for the treatment of diabetes and obesity, and new, highly specific therapeutic PTP1B inhibitors continue to be explored [16,17]. In this review, we provide an up-to-date summary of the molecular role of PTP1B in regulating immune cell function and how targeting its expression and/or activity has the potential to change the outcome of immune-mediated diseases.

## 2. Role of PTP1B in Haematopoiesis and Immune Cell Development

PTP1B is a ubiquitously expressed non-receptor tyrosine phosphatase encoded by the gene *ptpn1*. It comprises a highly conserved N-terminal catalytic domain, which contains cysteine residues involved in dephosphorylating tyrosine residues, two proline-rich motifs and an oestrogen-targeting domain on its C-terminus [2,18]. PTP1B has been implicated in controlling most cells of the innate immune system, from their development in haematopoiesis to their functional roles both in homeostasis and disease. During post-natal haematopoiesis, multipotent hematopoietic stem cells (HSCs) residing in the bone marrow commit to either the myeloid or lymphoid lineage. Common myeloid progenitors give rise to erythrocytes, platelets and granulocytes/monocytes, while common lymphoid progenitors give rise to T and B lymphocytes and natural killer cells, and both have potential for dendritic cell differentiation [19]. PTP1B is expressed in the early development of haematopoietic cells and has been shown to dephosphorylate activated JAK2, Tyk2 and STAT5 to control downstream signalling and differentiation [13,20]. PTP1B also regulates the phosphorylation levels of activated tyrosine kinase receptors controlling haematopoiesis, including the granulocyte colony-stimulating factor receptor (G-CSFR), colony-stimulating factor 1 receptor (CSF1R) and FMS-like tyrosine 3 ligand (FLT3), which modulate monocyte development and differentiation [21,22,23]. CSF1/CSF1R signalling, specifically, promotes the differentiation of myeloid precursors into monocytes, macrophages, dendritic cells and osteoclasts [24]. PTP1B-knockout mice have an increased ratio of monocytes to granulocytes in the spleen and blood, likely due to sustained activation of CSF1R in myeloid precursors, promoting monocyte differentiation [19,22,23,25]. Moreover, they show increased numbers of macrophage precursors, splenic monocytes and granulocytes due to their reduced apoptosis caused by changes in the expression of pro-apoptotic factors [19,22,23,25]. FLT3 is a receptor tyrosine kinase expressed by immature hematopoietic cells controlling the survival, proliferation and differentiation of haematopoietic and lymphoid progenitors [14,19,20]. Both CSF1R and FLT3 can signal through STAT3, the phosphorylation of which is controlled by PTP1B activity [26,27]. Leukaemia results from the dysregulation of haematopoiesis, leading to immature or blast cells and the deletion of chromosome 20q. Mutations in PTP1B, which is located on the human chromosome 20q13.1-q13.2, is associated with acute myeloid leukaemia and myeloproliferative neoplasms due to uncontrolled haematopoiesis [28]. Aged mice where PTP1B is knocked out in myeloid cells show a greater susceptibility to acute myeloid leukaemia, again highlighting the importance of PTP1B in myelopoiesis. PTP1B is also implicated in regulating granulopoiesis and PTP1B knockout in induced pluripotent stem cell (iPSC)-derived neutrophils, which are less mature than wild-type ones, with a lower expression of CD16 [29]. Allergen-challenged PTP1B-knockout mice have increased numbers of eosinophil precursors in their bone marrow as well as increased numbers of circulating and migrating eosinophil progenitors, establishing that PTP1B also influences eosinophil development [25]. PTP1B regulates B cell lymphopoiesis by dephosphorylating the FLT3 receptor, impairing downstream signalling and disrupting cell development, differentiation and homeostasis [21,27,30]. Consequently, PTP1B-null mice have increased numbers of immature B cells in their bone marrow, lymph nodes and blood, along with decreased B cell apoptosis in the lymph nodes, resulting in lymphomas [31]. T cell development, proliferation and survival are also regulated by PTP1B via dephosphorylating STAT5 [32]. Thus, PTP1B expression or activity is involved in the “poiesis” of several immune cell types, which is, in part, due to variations in the phosphorylation levels of receptor tyrosine kinases.

## 3. Role of PTP1B in Innate Immune Cell Responses

The innate immune system provides the first line of defence against invading pathogens, cancer cells, toxins or tissue damage. An important function of innate immunity includes the rapid recruitment of immune cells to sites of infection and inflammation through the production of cytokines and chemokines. These cells include neutrophils, monocytes/macrophages and mast cells, as well as dendritic cells, as a link between innate and adaptive immunity. The importance of PTP1B in driving the functional characteristics of innate immune cells is outlined below (Figure 1).

### 3.1. Neutrophils

Neutrophils are the most abundant white blood cells and have a short half-life, which is reflected in the release of billions of them from the bone marrow each day. Neutrophils are usually the first responders during the acute phase of inflammation. They are professional phagocytes, recognising and clearing pathogens such as bacteria and fungi as well as destroying microbes and tissue through degranulation and reactive oxygen species (ROS) production. Neutrophils also produce neutrophil extracellular traps (NETs) that trap and kill extracellular microbes and release anti-microbial peptides. These defence mechanisms become less potent as neutrophils age. Indeed, PTP1B inhibition has been shown to speed up neutrophil ageing in mice by reducing the activation of the PI3Kγ/AKT/mTOR signalling pathway, which downregulates CD62L and CXCR2 expression, markers of newly released neutrophils, as well as upregulating CXCR4 signalling, which clears aged neutrophils from the blood [33]. Overall, PTP1B inhibition significantly affects neutrophil function by decreasing their overall granule contents, maturity and ability to form NETs [33]. Neutrophils contain neutrophil elastase, a serine protease involved in NET formation and promoting inflammation through the secretion of pro-inflammatory cytokines. These cytokines upregulate PTP1B expression, and there is a correlation between the expression of neutrophil elastase and PTP1B [33,46,47,48]. Inhibiting neutrophil elastase has anti-inflammatory effects and prevents the formation of NETs, improving atherosclerotic plaque formation and reducing lung injury in mice [49,50,51]. Thus, neutrophil elastase inhibition could potentially decrease PTP1B expression and dampen inflammatory cytokine production [33]. It was reported that PTP1B-knockout iPSC-derived neutrophils led to enhanced migration via increased phosphorylation of ERK1/2 and hematopoietic lineage cell-specific protein 1 (HS1), which regulate actin dynamics during cell migration [29]. These iPSC neutrophils showed improved phagocytosis compared to wild-type neutrophils but decreased their production of ROS and NETs, consistent with other studies [29,33]. The improved phagocytic ability of PTP1B-deficient neutrophils is due, at least in part, to the upregulated expression of FcɣR1, which clears antibody-opsonised microbes [36]. This was confirmed in the presence of *Aspergillus fumigatus*, where PTP1B-knockout iPSC neutrophils were found to be highly motile and phagocytic for the fungus compared to wild-type neutrophils [29]. Similarly, neutrophil clearance of *Pseudomonas aeruginosa* is impaired by the enhanced PTP1B expression observed with infection through negatively regulating the Toll-like receptor 4 (signal transducer and activator of transcription factor 1)-inducible nitric oxide synthase (TLR4-STAT1-iNOS) signalling pathway and nitric oxide (NO)-mediated bacterial killing [36]. PTP1B-deficient neutrophils also enhance bacterial clearance in pulmonary *Pseudomonas aeruginosa* infection by upregulating interferon regulatory pathways and cytokine production [36]. Moreover, inflammatory cytokines such as IL-6 and tumour necrosis factor α (TNFα) were also found to be upregulated in PTP1B-knockout iPSC neutrophils upon lipopolysaccharide (LPS) stimulation, and this may aid in neutrophil defence [29].

### 3.2. Macrophages 

Macrophages play key roles in innate immunity by phagocytosing microbes and cellular debris, producing inflammatory cytokines to activate other cell types and restoring tissue homeostasis after injury. Macrophages exhibit such varied functions due to their heterogeneity [22,52]. In response to microenvironmental stimuli, macrophages polarise into a range of functional phenotypes, with the extremes being M1-like pro-inflammatory cells that drive inflammation, pathogen and tumour clearance and M2-like macrophages that have anti-inflammatory, tissue-reparative and immunosuppressive functions but can promote tumourigenesis [53]. Monocytes/macrophages express relatively high levels of PTP1B, and this increases upon activation with pro-inflammatory stimuli such as cytokines, LPS, saturated fatty acids or ionising radiation [39,54].

PTP1B is key in regulating the expression, secretion and responses of both pro- and anti-inflammatory cytokines in human and murine macrophages [39,55]. The inhibition of PTP1B results in upregulated phosphorylation of STAT1/3, JNKs, p38, ERK/MAPKs, NF-κB and p65 (MyD88-dependent and TRIF-dependent pathway activation) and increased production of IL-10, TNF, IL-6 and interferon (IFN)-β in LPS/palmitate-activated macrophages [39,55,56]. PTP1B is a well-known negative regulator of type I IFN signalling in several cell types, including macrophages, and PTP1B silencing can enhance IFN secretion and help in the clearance of bacteria and viruses through increased STAT1 phosphorylation [12,22,37]. However, it has also been reported that PTP1B-knockout macrophages secrete lower amounts of type I IFN than cells from wild-type mice when activated through the pattern recognition receptors cGAS, RIG-1, TLR7 and TLR9, and this is independent of PTP1B phosphatase activity, suggesting that these effects are related to the types of activation, the environment and the cells involved [57]. Tissue macrophages lacking PTP1B exhibit increased CD80 expression, which is associated with inflammatory responses and the activation of the adaptive immune system [22]. PTP1B strongly influences macrophage M1- or M2-like polarisation and downstream functions depending on the stimulus and experimental models and whether these models involve global or cell-specific PTP1B knockout [22,39,58,59]. PTP1B has been shown to suppress IL-4 macrophage activation [58]. IL-4 signals via PI3K to promote PTP1B expression and stability; however, PTP1B then feeds back to attenuate IL-4-induced STAT6 phosphorylation by dephosphorylating STAT6, thus preventing M2-like polarisation [58]. Consistent with this, murine PTP1B-knockout macrophages displayed a more anti-inflammatory, M2-like phenotype by enhancing IL-10-dependent STAT3 phosphorylation and suppressing LPS-induced M1-like macrophage activation [59]. Thus, PTP1B and its inhibition control both M1- and M2-like activation through effects on specific polarisation signalling pathways [58,59].

PTP1B expression is also important for macrophage metabolic activity and downstream viability [39]. Following 5G ɣ-irradiation, macrophages from PTP1B-deficient mice showed enhanced sensitivity to ionising radiation [38,39]. Animals lacking PTP1B displayed decreased metabolic activity after sub-lethal ɣ-irradiation, and PTP1B-deficient macrophages had a lesser capacity to regulate oxidative stress and phagocytosis due to decreased expression of the pro-survival gene BCL-2 and increased expression of the pro-apoptotic protein BAX [39].

### 3.3. Mast Cells 

Mast cells play a crucial role in allergic inflammation as they contain abundant cytoplasmic granules, such as histamine, as well as proteolytic enzymes that kill bacteria. They also synthesise and secrete lipid mediators and cytokines which stimulate inflammation. Indeed, the crosslinking of the FcεR1 on mast cells with an antigen leads to the activation of multiple signalling pathways, including MAPK, Akt and NF-κB, and the release of inflammatory mediators [40,41].

However, little is known about the exact role of PTP1B in mast cells. It has been reported that FcεR1 crosslinking leads to PTP1B and Syk interaction, resulting in mast cell activation through the attenuation of the inhibitory AMPK-mediated activation pathway and conversely enhanced stimulatory Syk pathway [40]. PTP1B deficiency in mast cells heightened IgE-mediated STAT5 phosphorylation, which in turn led to an increase in CCL9 and IL-6 expression [41]. Mast cell degranulation and cutaneous reactions were not affected by PTP1B deficiency in vivo, suggesting no impact on mast cell effector function [41]. Thus, the effects of PTP1B on mast cells are most likely to be secondary effects mediated through signalling effects by Fc receptors.

### 3.4. Dendritic Cells (DCs)

Dendritic cells (DCs) are efficient antigen-presenting cells that activate T cells and are therefore crucial for linking innate and the adaptive immune responses. Antigen processing leads to DC maturation, which is regulated by the JAK/STAT pathway and characterised by the upregulation of chemokine receptors such as CCR7 and co-stimulatory molecules such as CD80/CD86 and the expression of major histocompatibility complex (MHC) class I and II molecules required for presenting antigens to CD4^+^ and CD8+T cells.

Recent studies have shown PTP1B to have an essential role in regulating several stages of DC immune responses, such as their maturation, migration to lymph nodes and T cell activation [42]. The genetic deletion of PTP1B in murine myeloid cells impaired LPS-driven bone marrow-derived DC activation by increasing STAT3 phosphorylation [42]. These DCs exhibited decreased expression of CCR7 on the cell surface, and they efficiently migrated to lymph nodes despite the presence of CCL19, a chemokine important for homing DCs to lymph nodes [2]. Similarly, mice with the total deletion of PTP1B expression developed tolerogenic DCs with decreased MHC class II expression and an impaired ability to activate T cells due to STAT3 hyperphosphorylation [43]. However, the heterozygous deletion of PTP1B enhances the hyperactivation of STAT1 and STAT4 and the expression of MHC class I, IFN-ɣ and IL-12, essential for DC-mediated T cell activation and DC maturation, thus giving rise to immunogenic DC [43]. DC maturation is therefore regulated by PTP1B expression levels, which consequently affects DC activity [43].

PTP1B is required for the fully functional activation of T cells by DCs [42]. Indeed, PTP1B deletion in DCs led to a 50% decrease in DC-to-T cell contact due to the low expression of co-stimulatory molecules (CD80 and CD86) and MHC class I expression, suggesting that PTP1B is required for successful antigen presentation to T cells and their subsequent activation [42].

## 4. Role of PTP1B in Adaptive Immune Responses

The adaptive immune system encompasses more specific recognition of non-self-antigens, which involves a highly regulated interaction between loaded antigen-presenting cells and antigen-specific receptors expressed on lymphocytes. The main cells in the adaptive immune system are T and B cells, which carry out cell-mediated immunity and antibody responses, respectively. The importance of PTP1B in regulating signalling in T and B cells and their downstream activation and function are outlined below.

### 4.1. Activated B Lymphocytes 

Activated B lymphocytes are involved in the humoral response through secreted antibodies. Upon antigen encounter, B cell receptors expressed on the cell surface mediate the antigen-specific activation of B cells, leading to their proliferation and differentiation into antibody-secreting plasma cells. As mentioned previously, PTP1B regulates B cell development but is also involved in B cell receptor (BCR) signalling and consequently B cell activation [31]. B cell activation is regulated by co-receptors such as CD40, and B cell activating factor receptor (BAFF-R), involved in cell survival and PTP1B, negatively regulates CD40, BAFF-R, and Toll-like receptor 4 (TLR4) signalling in B cells by dephosphorylating MAPK and p38 [44].

CD40 signalling in B cells is involved in the interaction with CD4^+^ T cells, and a switch to high-affinity IgG antibody production and PTP1B can influence this [44]. Anti-CD40 and BAFF stimulation led to a significantly higher proliferation rate and survival of PTP1B-deficient B cells compared to B cells expressing PTP1B, and these knockout mice developed systemic autoimmunity and were more susceptible to T cell-mediated immune activation [44]. In addition, the expression of transcription factors such as SYK, ERK, BTK and PLCɣ2, which are involved in B cell activation, was increased in PTP1B-knockout B cell lines [45]. Overall, this confirms that PTP1B negatively regulates B cell activation and function.

### 4.2. T Lymphocytes 

T lymphocytes are involved in cell-mediated immunity. There are two main types of T cells, cytotoxic CD8^+^ T cells and helper CD4^+^ T cells, which play distinct roles in the immune system. While CD8^+^ T cells kill infected or tumour cells, CD4^+^ T cells secrete cytokines and other mediators to coordinate and direct the immune response.

It is well established that IL-2/5/15-induced STAT5 signalling is essential for T cell development, activation, proliferation, homeostasis and survival and that PTP1B attenuates JAK/STAT signalling pathways. However, T cells deficient in PTP1B do not develop systemic inflammation or autoimmunity [35]. Instead, PTP1B has been highlighted as a negative regulator of T cell differentiation, as T cell-specific PTP1B deletion in mice led to an increase in peripheral naïve, effector/memory and central memory T cells in lymphoid organs [35]. PTP1B expression is increased in murine intratumoural CD8^+^ T cells, repressing antitumour immunity. Inhibiting PTP1B expression in T cells has been shown to limit the growth of implanted AT3-OVA mammary tumours due to the increased infiltration of effector/memory CD4^+^ and CD8^+^ T cells, promoting T cell-mediated antitumour immunity related to enhanced STAT5 activity [32,35]. Thus, PTP1B is now identified as a clinical immune checkpoint playing a critical role in the treatment of tumours by inhibiting the proliferation and cytotoxicity of T cells induced by tumour antigen [32,35]. The pharmacologic inhibition of PTP1B enhanced the response to PD-1 checkpoint blockade by repressing tumour development and decreasing T cell exhaustion, and it also promoted chimeric antigen receptor (CAR) T cell-mediated antitumour immunity, highlighting PTP1B inhibition as a promising target for cancer immunotherapy [34,35].

## 5. The Role of PTP1B in Disease

It has been established that PTP1B expression and activity are upregulated following the activation of several immune cell types, and this is important in modulating signalling pathways and downstream functions. The dysregulation of PTP1B expression or activity can have a profound impact on cell signalling and disrupt the immune response by directly affecting immune cell development, activation and function. This changed expression or activation has been shown to lead to malignancy, chronic inflammatory pathologies, susceptibility to infection and metabolic or autoimmune diseases such as type 2 diabetes. Thus, PTP1B has potential as a therapeutic target for several conditions (Figure 2). Here, we will discuss selected inflammatory and autoimmune diseases as well as infectious diseases in which the PTP1B-induced modulation of cell function plays a significant role.

### 5.1. PTP1B in Inflammatory Diseases

Inflammation is a biological defence mechanism whereby the immune system responds to harmful or foreign stimuli and instigates a resolution process. Inflammation can be acute, with rapid responses to tissue damage that may last up to 6 weeks, e.g., acute pneumonia, or chronic, lasting for prolonged periods of several months to years, e.g., allergies. A hallmark of inflammation involves the secretion of pro-inflammatory mediators, such as cytokines and chemokines, and the recruitment and activation of immune cells that can eliminate harmful or foreign agents but damage host tissue, followed by a phase to try and resolve tissue damage through changing immune cell function to produce anti-inflammatory, immunoregulatory molecules [71]. A fine balance between pro- and anti-inflammatory mediators is essential to avoid uncontrolled inflammation. As PTP1B is a key regulator of immune cell activation, it is not surprising that PTP1B is upregulated in inflammation and has a role in controlling the immune response in chronic inflammatory diseases, as discussed below.

#### 5.1.1. Colitis

Myeloid-derived suppressor cells play a protective role in suppressing inflammation and autoimmunity. As highlighted previously, PTP1B is a negative regulator of myeloid cell production from bone marrow cells. In a model of murine experimental colitis, PTP1B deficiency was shown to promote the expansion of myeloid-derived suppressor cells in the bone marrow, blood, and spleen by activating the JAK2/STAT3 pathway. This decreases neutrophil infiltration and IL-17 levels associated with colitis, therefore improving the outcomes of diseases [60].

#### 5.1.2. Asthma and Allergic Inflammation

In the context of asthma, exposure of pre-sensitised animals to allergens, e.g., ovalbumin, typically results in increased recruitment of inflammatory cells, such as lymphocytes and eosinophils, to the lungs [61]. PTP1B has been found to be a negative regulator of allergic inflammation, limiting allergic responses. PTP1B-deficient mice showed increased leukocyte recruitment, particularly eosinophils, following ovalbumin sensitisation, establishing that PTP1B is important in regulating the early migration and chemotaxis of inflammatory cells [25]. The levels of the Th2 cytokines IL-4, IL-5 and IL-10 were also found to be decreased following sensitisation with ovalbumin in PTP1B-knockout mice, confirming the importance of PTP1B in regulating large-scale inflammation [62].

#### 5.1.3. Atherosclerosis

Chronic inflammation is a major factor in the development of atherosclerosis, which is driven by the activation of pro-inflammatory macrophages responsible for atherogenesis and plaque formation through the secretion of inflammatory mediators and matrix-degrading enzymes. On the other hand, anti-inflammatory macrophages promote the resolution of atherosclerosis as well as controlling cholesterol efflux and efferocytosis [72]. Our own studies have demonstrated that the myeloid-specific genetic deletion of PTP1B in an ApoE^−/−^ mouse model of atherosclerosis protected against atherosclerosis development by increasing the circulating levels of the anti-inflammatory IL-10 cytokines and the vasodilator PGE_2_, along with decreasing the circulating levels of cholesterol and triglycerides, defending against atherosclerotic plaque formation [63]. Moreover, the pharmacological inhibition of PTP1B with trodusquemine (MSI1436) prevented atherosclerotic plaque formation in an LDLR^−/−^ mouse model of atherosclerosis. This was due to reduced circulating total cholesterol and triglycerides, along with a decrease in aortic monocyte chemoattractant protein-1 (MCP-1) expression levels and the hyperphosphorylation of aortic Akt/PKB and AMPKα [63]. PTP1B is known to regulate oxidative stress, an effect prominent in atherosclerosis, and cells deficient in PTP1B show reduced oxidative stress upon activation by preventing glutathione reduction, the generation of ROS and the activation of JNKs and p38-mitogen-activated protein kinases (p38-MAPKs) [64].

Further studies on human cells showed that pharmacological PTP1B inhibition decreased macrophage foam cell formation by increasing cholesterol efflux to high-density lipoproteins through AMPK activation [73]. Overall, PTP1B potentially promotes atherosclerosis by enhancing a macrophage pro-inflammatory phenotype, regulating oxidative stress, altering the levels of circulating lipoproteins in an IL-10-dependent manner and dysregulating cholesterol efflux in macrophages [73,74]. PTP1B is also well known to regulate diabetes and metabolic syndrome, which accelerate atherogenesis [75].

#### 5.1.4. Sepsis

Sepsis is triggered by the dysregulation of the host inflammatory response, leading to systemic immune responses and then to infections. Neutrophils and macrophages are the major types of immune cells involved in this disease and have both been found to have increased production of ROS along with diminished secretion of pro-inflammatory cytokines by macrophages [76]. Consistent with this, PTP1B inhibition improves survival in LPS-induced sepsis by accelerating neutrophil ageing to respond to inflammatory stimuli [33,49,65]. This suggests that PTP1B inhibitors are protective against sepsis models by dampening the cytotoxic nature of neutrophils [33]. It is of interest that patients with septic shock have a *PTPN1* gene expression variation which is partly correlated with the evolution of septic organ failure [49].

#### 5.1.5. Alcoholic Liver Disease

PTP1B has multiple pharmacological roles in liver disease, such as alcohol liver injury, where PTP1B mRNA and protein levels were shown to be increased in the liver of ethanol-fed mice and in macrophages isolated from the hepatic tissue of these mice [66]. Alcohol-stimulated macrophages secrete pro-inflammatory cytokines such as IL-1β and TNFα, which promote the development of alcoholic liver injury [66]. PTP1B was shown to have a dual role in modulating inflammation in alcoholic liver injury [77]. While PTP1B acts as a negative regulator of TLR signalling by preventing MyD88- and TRIF-dependent production of pro-inflammatory cytokines, PTP1B knockout has been shown to decrease the phosphorylation levels of NF-κB in alcohol-stimulated macrophages, which modulates the expression of many inflammatory genes involved in promoting alcohol liver injury [66,77]. PTP1B pharmacological inhibition procured beneficial effects and mitigated hepatic injury, inflammation and steatosis caused by ethanol feeding [78,79].

### 5.2. PTP1B in Autoimmune Diseases

PTP1B expression and activity have been implicated in several autoimmune diseases, such as diabetes, rheumatoid arthritis (RA) and neurodegenerative disorders. This relates to the role of PTP1B in macrophage activation, T and B cell differentiation and activation, antibody production, cytokine signalling and inflammation [42].

#### 5.2.1. Diabetes

Inflammation is the predominant event in type 1 diabetes, where infectious and/or autoimmune processes initiate disease, while chronic inflammation is common in type 2 diabetes due to increased insulin resistance and disturbed glucose metabolism. The role of PTP1B in negatively regulating the insulin and leptin signalling pathways and the downstream effects on glucose homeostasis and body mass, which result in metabolic disorders such as type 2 diabetes and obesity, are well established [80,81]. A number of immune cells, such as eosinophils, macrophages and T regulatory cells, have been found to be present in the adipose tissue of people with type 1 and 2 diabetes, with the role of maintaining homeostasis by secreting cytokines, the expression of which is controlled by PTP1B [82]. Due to its role in negatively regulating insulin signalling, PTP1B expression can result in high blood glucose levels, potentially acting as a metabolic trigger to immune cells, driving further inflammation and inflammaging. It has been shown that myeloid cell-specific PTP1B knockout in long-term high-sugar-diet-fed mice promotes an anti-inflammatory phenotype via the STAT3-dependent expansion of IL-10-secreting macrophages while downregulating TNFα expression, thus preventing metabolic ageing [65]. Since IL-10 has been extensively shown to have an insulin-sensitising effect, it has been found that PTP1B deletion in myeloid cells has a protective role in the development of diabetes by alleviating hepatic and adipose inflammation [65,80]. After a number of years, poorly controlled diabetes instigates downstream disorders such as diabetes-induced non-healing wounds and diabetic retinopathy and nephropathy [83,84]. PTP1B has been highlighted as having a role in controlling diabetic wound healing and diabetic nephropathy through effects on immune cells [54,73,84,85].

#### 5.2.2. Rheumatoid Arthritis (RA)

Rheumatoid arthritis is a systemic autoimmune disease in which both the innate and adaptive immune systems are involved, leading to unusually high levels of inflammation [65]. Inflammatory cells such as macrophages have an elevated expression of PTP1B, which relates to their inflammatory potential, suggesting PTP1B inhibition could be therapeutic for this condition [39,86]. The production of autoantibodies secreted by autoreactive B cells is also pivotal in inflammation and the pathogenesis of RA. By controlling B cell receptor downstream signalling, PTP1B negatively regulates the B cell expression of CD40, B cell-activating factor receptor (BAFF-R) and TLR4 signalling. Therefore, the deletion of PTP1B specifically in B cells led to an acceleration of autoimmune responses in mice. In human patients with RA, B cells exhibited reduced expression of PTP1B. This reduction was associated with heightened p38 MAPK activity, resulting from PTP1B’s impaired ability to dephosphorylate receptor-associated kinases. Consequently, this increased the inflammatory response characteristic of RA [44]. Thus, pharmacologically targeting PTP1B in patients with RA could have dual effects: it might reduce inflammation by modulating macrophage activity, but at the same time, it could potentially increase inflammation driven by B cells [44].

#### 5.2.3. Neurodegenerative Diseases

Neuroinflammation is a common characteristic in neurodegenerative disorders, and PTP1B expression has been found to be upregulated in inflammatory brain conditions. As such, PTP1B inhibition in LPS-stimulated microglial cells suppressed the expression of pro-inflammatory cytokines such as TNFα, iNOS and IL-1β via Src-mediated phosphorylation, suggesting the potential of PTP1B as a therapeutic target for neuroinflammation and neurodegenerative diseases, including Alzheimer’s disease, Parkinson’s disease and multiple sclerosis (MS) [67]. Indeed, the inflammatory-mediated activation of PTP1B has been shown to disrupt metabolic-related signalling pathways in Alzheimer’s disease, leading to disease progression [69]. PTP1B inhibition has been shown to decrease inflammation in the hippocampus of mice by decreasing microglial activation [69]. Similarly, PTP1B inhibition dampens inflammation in zebrafish models of Parkinson’s disease by reducing the expression of IFN-ɣ-induced pro-inflammatory cytokines such as iNOS, COX2 and NF-κB while upregulating M2 markers in microglial cells, improving neuronal damage [68]. PTP1B expression has also been found to be upregulated in autoimmune encephalomyelitis (EAE), an experimental model of brain inflammation and MS, disrupting tight junctions in the blood–brain barrier, which promotes immune cell infiltration, one of the main causes of MS in humans [87]. CD4^+^ T helper (Th) cells secreting IFN-ɣ and CD4^+^ Th17 cells secreting pro-inflammatory cytokines such as IL-17 and IL-6 are thought to be the main mediators of MS [88]. Hence, the anti-inflammatory effects resulting from PTP1B inhibition could potentially have a protective role in neurodegenerative diseases by suppressing microglial hyperactivation and the consequent recruitment of immune cells to the brain [87,88,89].

### 5.3. PTP1B in Infectious Diseases

Infectious diseases are caused by a wide range of pathogens, with viruses, bacteria and fungi being the most prominent. The host counteracts these infections through immune cell defences, and this results, in many cases, in an enhanced expression of PTP1B, which controls their functional effects.

#### 5.3.1. Viral Infections

Pattern recognition receptors (PPRs) are required for the detection of invading pathogens and to induce an efficient immune response through the secretion of pro-inflammatory mediators [57]. It has been established that PTP1B plays a role in viral infections, such as cytomegalovirus, herpes simplex virus type 1, influenza A and respiratory syncytial virus, by regulating type I interferon signalling [37,57,90]. Interferon signalling pathways require the activation of the JAK/STAT1 or STAT2 pathway, which has been extensively shown to be negatively regulated by PTP1B [35,42]. In one study, PTP1B deficiency did not affect the signalling pathway but led to an accumulation of type I interferon in macrophages, suggesting that PTP1B has important roles in trafficking and secreting interferon independently of its phosphatase activity [57].

#### 5.3.2. Bacterial Infections

Toll-like receptors (TLRs) are an essential mechanism for driving host defence against microbial infections. Indeed, TLR signalling has been shown to be negatively regulated by PTP1B in *Pseudomonas aeruginosa* infections [91]. More specifically, PTP1B-deficient mice showed enhanced activation of IRF7 interferon-stimulated response elements, which upregulate the expression of chemokines such as CCL5, CXCL10 and IFN-β, promoting the clearance of *Pseudomonas aeruginosa* [91]. It has also been shown that PTP1B negatively regulates phagocytosis and the NO-dependent killing of *Pseudomonas aeruginosa* by neutrophils in a TLR–STAT1–iNOS-dependent manner, interacting specifically with STAT1 [36]. PTP1B has the same effect on macrophage phagocytosis, although this is STAT1-independent, suggesting that PTP1B inhibition protects from *Pseudomonas aeruginosa* infection by enhancing the recruitment, phagocytic ability and NO-mediated bactericidal killing of both neutrophils and macrophages [36,70].

#### 5.3.3. Fungal Infection

There is a lesser body of literature on the role of PTP1B in regulating immune cell responses to fungal infection. PTP1B-knockout iPSC neutrophils have a highly activated phenotype and are more responsive to the fungal pathogen *Aspergillus fumigatus* [29]. These PTP1B-knockout iPSC neutrophils migrated efficiently to and swarmed hyphae, resulting in a significant inhibition of fungal growth [29]. It is of interest that fungi, such as Aspergillus species, produce many secondary metabolites that can inhibit PTP1B activity, making them an attractive source for the isolation of naturally available PTP1B inhibitors [92,93]. The conditions by which these metabolites are produced by Aspergillus are still unclear but have been linked to changes in nutrient availability or environmental stress [94]. Future studies are needed to define the specific effect of these Aspergillus-produced PTP1B inhibitors on the effectiveness of fungal clearance by immune cells.

## 6. Conclusions and Future Directions

Despite PTP1B being a relatively simple enzyme, it has the ability to dephosphorylate several substrates; thus, it is a critical protein in regulating many different cellular processes, including cell differentiation, activation, metabolism, proliferation, survival and migration. The role of PTP1B in regulating metabolic diseases and cancer is well established, drawing significant interest from pharmaceutical companies, some of which have developed PTP1B inhibitors that have progressed to clinical trials. PTP1B’s involvement in modulating signalling pathways that drive immune and inflammatory responses also highlights its potential as a therapeutic target for immune-mediated and inflammatory disorders where there is dysregulation. This therapeutic potential includes the ability to either reduce inflammation or enhance immune responses, potentially improving the efficacy of treatments such as PD-1/PD-L1 blockade therapy or CAR T cell therapy for cancer [32,35]. Therefore, PTP1B inhibitors not only play a crucial role in controlling cancer cell growth directly but also offer the added benefit of enhancing T cell responses to target and kill cancer cells more effectively. Given the wide variety of substrates that PTP1B interacts with, its regulatory functions are highly dependent on the cellular context, the specific protein substrates and the overall metabolic state of the cells. For instance, PTP1B clearly inhibits macrophage activation yet shows conflicting results regarding its role in macrophage phenotype determination, with PTP1B inhibition leading to both M1-like and M2-like phenotypes in different contexts. Similarly, PTP1B was shown to negatively regulate B cell activation and impair BCR re-arrangements in mice, but the opposite was shown when using B cell lines [44]. Additionally, while PTP1B is necessary for mast cell activation, it acts as a negative regulator of allergic responses [25,40,41,62]. The development of specific PTP1B inhibitors or activators holds significant promise for both pharmaceutical and academic advancements. However, it is essential to thoroughly understand a range of PTP1B substrates and to identify their potential off-target effects or negative consequences on immune cell activation in various disease contexts. It is also critical to continue to improve the specificity of inhibitors and ensure efficient bioavailability. Once these aspects are clarified, the outcomes of manipulating PTP1B activity in health and disease will be more predictable, allowing for more targeted and effective therapeutic interventions.

## Figures and Tables

**Figure 1 ijms-25-07207-f001:**
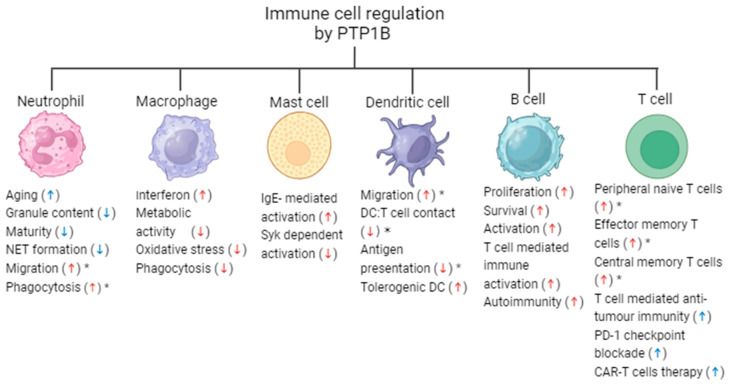
The role of PTP1B in immune cell function. PTP1B regulates the activation and functional properties of both innate and adaptive immune cells. The pharmacological inhibition (↑, ↓) [32,33,34,35] or knockout (↑, ↓) [12,22,29,35,36,37,38,39,40,41,42,43,44,45] of PTP1B can alter their functional properties, as shown by the arrows. The results refer to global knockout unless specified by an asterisk (*), which represents cell-specific knockout.

**Figure 2 ijms-25-07207-f002:**
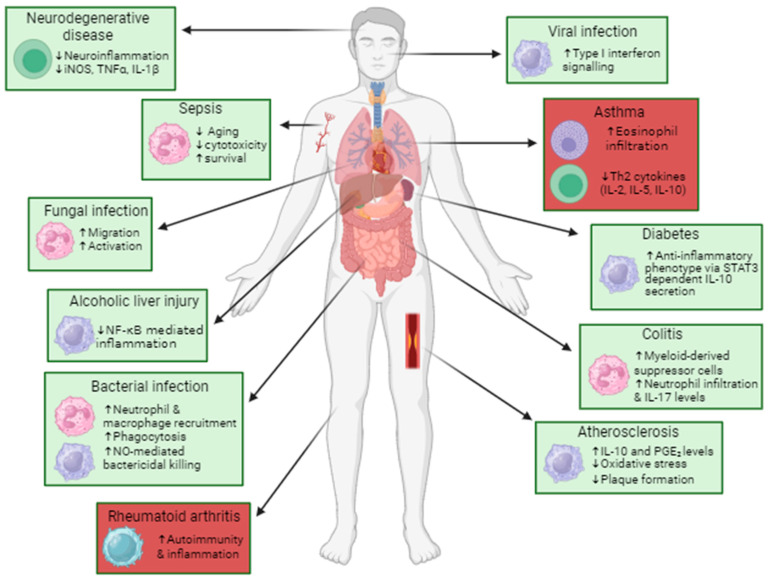
Therapeutic potential of PTP1B inhibition in immune-mediated diseases. Inhibiting PTP1B in immune-mediated diseases could have potential therapeutic effects. It can improve the outcome of disease (in green) by suppressing the pro-inflammatory phenotype of immune cells in autoimmune diseases, or it can help recruit immune cells to the site of infection and promote their activation. This, however, can be harmful in the context of other diseases (in red), exacerbating inflammation and promoting disease progression [25,29,33,36,44,49,57,60,61,62,63,64,65,66,67,68,69,70].

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
