# Peer review of "Recent Developments in the Role of Protein Tyrosine Phosphatase 1B (PTP1B) as a Regulator of Immune Cell Signalling in Health and Disease"

_ijms, 2024, doi:10.3390/ijms25137207_

Round 1
Reviewer 1 Report
Comments and Suggestions for Authors
comments are attached

Author Response
Dear Reviewer,
Thank you for the opportunity to revise our review based on your helpful comments. We consider these improve the review overall and we hope you now find it acceptable for publication. Below is a point-to-point response on how we addressed the individual comments. The revisions made to the submitted manuscript have been highlighted in yellow to assist in identifying the changes.
Comment 1 As you mention that “The role of PTP1B in regulating metabolic diseases and cancer has been well established, attracting attention from pharmaceutical companies to develop PTP1B inhibitors that have reached clinical trials”, could you please provide a list of PTP1B inhibitors, as I believe there are currently no commercially available drugs that specifically target PTP1B inhibition. What are the primary factors contributing to this absence? There are numerous studies accessible in the literature that discuss inhibitors, ranging from those of natural origin to those that are synthetic. It would be beneficial to emphasize the primary factors and challenges in this particular situation.
Response 1: We thank the reviewer for this comment. A recent review by Delibegovic et al, listed a number of naturally derived and synthetic PTP1B inhibitors (small molecules, allosteric molecules, peptide inhibitors, antisense therapeutics) that have been developed over the years which have shown promising results in vitro. We have quoted this up to date review in our manuscript and only listed a few inhibitors in our manuscript due to word limits. Several lead drug candidates targeting PTP1B have shown promising effects and reached phase II clinical trials for T2DM and/or obesity, such as ertiprotafib, TTP-814, IONIS 113715 and IONIS PTP1BRx. Some have reached phase I trials (trodusquemine and KQ-791) and others are being investigated in pre-clinical studies (JTT-551, DPM-1001 and DPM-1003). Unfortunately, most of them did not succeed in subsequent clinical trials due to the highly conserved catalytic site of PTP1B with other PTPs, such as TC-PTP, a lack of physiological effects or safety concerns. These points with an emphasize the primary factors and challenges have now been included in the revised manuscript
Comment 2: What potential impact might a neutrophil elastase inhibitor have on PTP1b in the context of inflammation? If there is any way in which such an inhibitor could have a beneficial effect on PTP1B inhibition, highlight such details.
Response 2: Neutrophil elastase has been shown to be involved in NET formation and in promoting inflammation. Song et al., has shown that PTP1B inhibitors impaired NETosis by decreasing MPO containing primary granules. MPO are responsible for the translocation of neutrophil elastase to the nucleus and decondensation of chromatin. Inhibiting neutrophil elastase promotes anti-inflammatory effects and, consequently, could dampen PTP1 expression. This point has now been included in the manuscript.
Comment 3: The role of PTP1B is unclear in the case of Rheumatoid arthritis. What does it represent lower expression, and then targeting, elaborate this well.
Response 3: Due to the different types of immune cells involved in RA, PTP1B inhibition in could have a dual effect. The presence of highly inflammatory immune cells such as macrophages could be beneficial for the resolution of RA-induced inflammation following PTP1B inhibition as it has been shown that PTP1B inhibition promotes an anti-inflammatory phenotype in macrophages. However, B cells isolated form RA patients have lower expression of PTP1B, which is unable to dephosphorylate p38 MAPK, increasing inflammation. Hence further inhibition of PTP1B could leads to sustained P38 MAPK signalling and inflammation. This section has now been rewritten for clarity.
Comment 4: In respect to previous articles what are the novelty present in current review article, for example can the author highlight some comparison, I can mention here, “Recent advances in understanding the role of protein-tyrosine phosphatases in development and disease”. By Hale et.al, Also the authors can see “Protein Tyrosine Phosphatase 1B (PTP1B) in the immune system” by Samantha et.al.
Response 4: Hale et al, (2017) focusses on many PTPs and does not cover the effects of PTP1B in immune cells and Le Sommer et al, is an old review (2015) which focusses more on disease rather than all immune cells. Here we provide up to date info on PTP1B specifically in immune cells and the potential downstream implications in disease. The uniqueness is the emphasis on immune cell functions specifically which has not been reviewed previously.
Comment 5: The similarity index seems very high, the authors should rewrite most of the sentences
Response 5: We have checked the similarity report and although this passed the similarity index when check by the Journal, we have now addressed this point and rewritten several sentences to reduce this.
Comment 6: In case of diabetes the authors can cite the following manuscript to well describe the insulin and leptin receptors in case of PTP1B inhibitors.
Response 6: We have cited the first article “https://doi.org/10.1021/acsomega.3c07471”.
Comment 7: Arrange the references in the text, for example in section 2.
Response 7: We found this comment unclear. References were arranged in the order they appear. We have added new references and checked the order of all. If the reviewer could specify what the comment refers to, we are happy to make any required adjustments.
Comment 8: Make uniform “CD4+ T” throughout the manuscript
Response 8: We have made the adjustments and made it uniform throughout the manuscript.
Comment 9: Reference 53, you repeated two times for one issue, one 53 should be removed.
Response 9: These are 2 different references. Reference 53 in the original manuscript (reference 66 in the revised manuscript) by Thompson, D, et al; is “Pharmacological Inhibition of Protein Tyrosine Phosphatase 1B Protects against Atherosclerotic Plaque Formation in the LDLR-/- Mouse Model of Atherosclerosis”. While reference 56 in the original manuscript (69 in the revised manuscript) by Thompson, D, et al; is “Myeloid Protein Tyrosine Phosphatase 1B (PTP1B) Deficiency Protects against Atherosclerotic Plaque Formation in the ApoE−/− Mouse Model of Atherosclerosis with Alterations in IL10/AMPKα Pathway”.
Comment 10: Section 7 should be rearranged
Response 10: The introductory part in section 7 has been changed to include neurodegenerative disorders generally instead of having the individual neurodegenerative diseases listed.

Reviewer 2 Report
Comments and Suggestions for Authors
Author Response
Dear Reviewer,
Thank you for the opportunity to revise our review based on your helpful comments. We consider these improve the review overall and we hope you now find it acceptable for publication. Below is a point-to-point response on how we addressed the individual comments. The revisions made to the submitted manuscript have been highlighted in yellow to assist the reviewers in identifying the changes.
Major points:
Comment 1: The reviewer suggests that the introduction can be expanded. It might be helpful if the authors can introduce briefly the classification of PTPs, for example how many different PTPs subtypes exist in cells and the difference between receptor vs non-receptor PTP. The authors listed the receptors that PTP1B targets. Are these receptors targeted specifically by PTP1B? Information on these topics can provide a big picture for the readers to understand the molecular role of PTP1B.
Response 1: Thank you for this comment. Many papers in the literature explain the different subtypes of PTP depending on the specificity to their target residues. Classical PTP specifically target tyrosine residues, while VH-1-like PTP target both tyrosine and threonine/serine residues. The first group can then be further subdivided into transmembrane receptor-like and non-transmembrane PTPs depending their cellular localisation which is dictated by their structure. These points have now been included in the introduction which has been expanded.
PTP1B has been shown to directly target EGFR, PDGFR, IR and IGFR which consequently regulates downstream signalling pathways such as PI3K/Akt/mTOR, RAS-MAPK and AMPK. We have modified the text for clarity
Comment 2: In the fungal infection section, the authors described that PTP1B knockout neutrophil is more active toward Aspergillus. However, many PTP1B inhibitors have been isolated from fungi. The reason why fungi produce PTP1B inhibitor that would potentially activate immune cells to kill the fungi is unknown so far. The reviewer thinks that the paragraph can be rephrased to present this point more clearly. The authors can also include more information on the following questions: is Aspergillus one of the fungi that produce those PTP1B inhibitors? And under what conditions do the fungi produce the PTP1B inhibitors?
Response 2: Three different Aspergillus strains have been found to produce secondary metabolites that have been identified to have PTP1B inhibitor function. The conditions under which Aspergillus produce PTP1B inhibitors is still unclear but nutrient availability, environmental factors and stress are potential confounding factors and could be involved in Aspergillus general defence mechanism and chemical communication. However, the role that these Aspergillus produced PTP1B inhibitors have in activating immune cell response to clear fungal infection is still unknown, further studies are required to elucidate this. The manuscript has been expanded to include these points.
Minor corrections:
Comment 1: The numbering for the sections needs to be adjusted. Sections 6,7,8 are subsections under section 5 logically. They should be numbered as sections 5.1, 5.2 and 5.3.
Response 1: We have numbered each section in the revised manuscript to assist the editors with the layout.
Comment 2: The font for the “diabetes” section is inconsistent with the rest of the text
Response 2: The font has been changed to make it consistent with the rest of the text.
Comment 3: The reviewer noticed a few grammatically errors through the manuscript. For example, in the abstract, “The function of PTP1B in the immune system was largely ignored before discovering that PTP1B negatively controls the Janus kinase – signal transducer and activator of transcription (JAK/STAT) signalling pathway which strongly modulates immune responses.”
Response 3: We have addressed the concerns of the reviewer and rewritten several sections of the manuscript to improve the grammar.

Round 2
Reviewer 1 Report
Comments and Suggestions for Authors
revised well